# Effects of Transcranial Direct Current Stimulation of Bilateral Supplementary Motor Area on the Lower Limb Motor Function in a Stroke Patient with Severe Motor Paralysis: A Case Study

**DOI:** 10.3390/brainsci12040452

**Published:** 2022-03-28

**Authors:** Sora Ohnishi, Naomichi Mizuta, Naruhito Hasui, Junji Taguchi, Tomoki Nakatani, Shu Morioka

**Affiliations:** 1Department of Therapy, Takarazuka Rehabilitation Hospital, Medical Corporation SHOWAKAI, 22-2 Tsurunoso, Takarazuka 665-0833, Japan; peace.pt1028@gmail.com (N.M.); pt1016@gmail.com (N.H.); taguchi@takara-reha.com (J.T.); ryouhoushi@gmail.com (T.N.); 2Department of Neurorehabilitation, Graduate School of Health Sciences, Kio University, 4-2-2 Umaminaka, Koryo, Kitakatsuragi-gun, Nara 635-0832, Japan; s.morioka@kio.ac.jp; 3Neurorehabilitation Research Center, Kio University, 4-2-2 Umaminaka, Koryo, Kitakatsuragi-gun, Nara 635-0832, Japan

**Keywords:** stroke, motor paralysis, supplementary motor area of the non-injured hemisphere, corticospinal tract excitability, transcranial direct current electrical stimulation, coherence, case report

## Abstract

In patients with severe motor paralysis, increasing the excitability of the supplementary motor area (SMA) in the non-injured hemisphere contributes to the recovery of lower limb motor function. However, the contribution of transcranial direct current stimulation (tDCS) over the SMA of the non-injured hemisphere in the recovery of lower limb motor function is unclear. This study aimed to examine the effects of tDCS on bilateral hemispheric SMA combined with assisted gait training. A post-stroke patient with severe motor paralysis participated in a retrospective AB design. Assisted gait training was performed only in period A and tDCS to the SMA of the bilateral hemisphere combined with assisted gait training (bi-tDCS) was performed in period B. Additionally, three conditions were performed for 20 min each in the intervals between the two periods: (1) assisted gait training only, (2) assisted gait training combined with tDCS to the SMA of the injured hemisphere, and (3) bi-tDCS. Measurements were muscle activity and beta-band intermuscular coherence (reflecting corticospinal tract excitability) of the vastus medialis muscle. The bi-tDCS immediately and longitudinally increased muscle activity and intermuscular coherence. We consider that bi-tDCS may be effective in recovering lower limb motor function in a patient with severe motor paralysis.

## 1. Introduction

Motor dysfunction of the lower limbs is recognized in many patients after a stroke. The severity of motor paralysis in the early post-stroke period affects the recovery of motor function in the lower limbs [1,2,3]. However, even in patients with severe motor paralysis, there are some patients in whom the motor function of the lower limb on the paretic side is more than proportional recovery [4,5]. Brain plasticity has been suggested to be related to variations in recovery [6,7]. Recently, with the development of neurophysiological techniques such as transcranial magnetic stimulation, diffusion tensor imaging, and functional magnetic resonance imaging (MRI), studies on the excitability of the corticospinal tract (CST) and motor-related areas in stroke patients have been conducted [6,7,8,9,10,11]. Additionally, coherence analysis of paired surface electromyography (EMG) recordings has suggested that common neural drive from motor-related areas to motor neurons can be quantitatively assessed during gait [12,13,14,15]. Coherence analysis measures linear correlations between pairs of signals in the frequency domain [16], and the beta frequency band is strongly associated with corticospinal drive [12,13,14,15,17].

To recover the motor function of the lower limb on the paretic side, it is important to increase the excitability of the motor-related areas of the injured hemisphere and the CST that output to the paretic lower limb muscles during movement of the paralyzed side [6,7,8,9,10,11]. However, there is a limited increase in the excitability of the affected hemispheric motor-related areas and CST in patients with severe motor paralysis [6,8].

Transcranial direct current electrical stimulation (tDCS) is a means to noninvasively excite the cerebral cortex and increase the excitability of motor-related areas and the CST [18,19,20,21]. It has been shown that tDCS to the primary motor cortex of the injured hemisphere increases the excitability of the CST and muscle strength of the paretic leg in stroke patients [18,22]. However, it is unknown how tDCS, including the non-injured hemisphere, affects the motor function of the lower limb on the paretic side. In particular, the activation of the supplementary motor area (SMA) in the non-injured hemisphere affects the recovery of the motor function of the lower limb on the paretic side in patients with severe motor paralysis [6,23].

Additionally, coherence in the beta band reflecting CST excitability is greatly reduced in post-spinal cord injury patients with severe motor paralysis, not only during voluntary movements but also during walking [24]. Furthermore, it has been shown that muscle activity of the lower limbs during walking decreases because of reduced CST excitability [25]. However, intervention methods to increase the excitability of the CST and muscle activity of the lower limbs during gait in patients with severe motor paralysis are insufficient. We hypothesize that tDCS to the injured hemisphere, as well as to the non-injured hemisphere SMA, increases the excitability of CST and the muscle activity of the lower limbs during gait in patients with severe motor paralysis. This study aimed to examine the immediate and longitudinal effects of assisted gait training using a long leg orthosis (KAFO) combined with tDCS on bilateral hemispheric SMA on the excitability of the CST and the muscle activity of the lower limbs during gait in a stroke patient with severe motor paralysis.

## 2. Materials and Methods

### 2.1. Participant

A post-stroke patient (80-year-old woman) with infarction of the left middle cerebral artery was admitted to the Takarazuka Rehabilitation Hospital 30 days after stroke onset. The patient was able to perform activities of daily living independently before the disease. The patient lived with her husband and her social role was that of a homemaker. MRI showed a high-signal response in a wide area centered on the corona radiata and the posterior limb of the internal capsule (Figure 1). At the time of admission, the patient had a 1/6 score on the Brunnstrom Recovery Stage of the lower extremity [26] and 0/22 score on the Fugl-Meyer Assessment [27] lower limb synergy item (FMS) [28,29], indicating severe motor paralysis of the right lower limb. Additionally, she scored 0/23 on the Trunk Impairment Scale (TIS), an assessment of trunk function [30]. The Functional Independence Measure (FIM) score for transfers was 1/7, indicating that the patient required total assistance in daily living activities. For each assessment, lower scores indicate negative results. Rehabilitation included physical, occupational, and speech therapies. The time spent on rehabilitation was 1 h/day (7 times/week) for each. This study was approved by the Ethics Committee of Takarazuka Rehabilitation Hospital (ethics review number: 20211005); written informed consent was obtained from the patient. CARE guidelines followed to ensure transparency in the case reporting.

### 2.2. Study Design

The study was a retrospective AB design. In period A, patients received only gait training using KAFO (Kawamura Gishi Inc., Osaka, Japan) with a therapist assisting from the back (assisted gait training; Figure 2). In period B, tDCS over SMA of the bilateral hemispheres combined with assisted walking training (bi-tDCS) was performed. Additionally, to examine the immediate effects of bi-tDCS on the motor function of the paretic lower limb, the patient was subjected to three conditions: (1) assisted gait training only (no-tDCS) for 20 min, (2) tDCS on the injured hemisphere SMA combined with assisted gait training (uni-tDCS) for 20 min, and (3) Bi-tDCS for 20 min in the intervals between periods A and B. The three conditions were performed on separate days, and the immediate effects were compared before and 20 min after training (Figure 3). The type of KAFO was an ankle joint with double klenzak and an oil damper, which has the function of resistance to the ankle plantarflexion movement by hydraulic pressure. The knee joint was a ring lock.

### 2.3. Setting up tDCS

The stimulation electrodes and sponge pads of the tDCS (DC-Stimulator Plus, NeuroConn, Ilmenau, Germany) were 5 cm × 7 cm (35 cm^2^), and the sponge pads were soaked with saline solution on their surfaces. A conductive gel was applied under the electrodes to reduce contact impedance. The anode position of the tDCS (Figure 4) was determined based on of the International Electroencephalogram 10–20 method, with the SMA of the injured hemisphere 3 cm anterior to the lateral side of the Cz and SMA of the bilateral hemispheres 3 cm anterior to Cz [31]. Under all conditions, the cathode was placed in the supraorbital region of the non-injured hemisphere, and the stimulation intensity was 2.0 mA for 20 min [18,20,22,32]. The current density was 0.057 mA/m^2^, which is within the safety guidelines for tDCS [33,34]. The ramp-up and ramp-down at the beginning and end of the stimulation were set to 10 s. After tDCS, the patient was verbally verified for adverse events and side effects [35].

### 2.4. Clinical Assessment and Measurement Items

For clinical assessment, BRS, FMS, TIS, and FIM scores for transfers at the period A end time (period B start time) and period B end time were measured. Wireless accelerometer, wireless surface EMG, and video data were recorded while walking. Wireless accelerometers (Gait Judge System: Pacific Supply Co., Ltd., Osaka, Japan; sampling rate: 1 kHz) were attached directly above the paretic lateral malleoli. Muscle activity in the proximal and distal portions of the vastus medialis (VM) muscle was measured during assisted gait training. We also assessed intermuscular coherence in the two paretic VMs as indicators of CST excitability. Previous studies have reported that CST controls the muscles of the thigh [8,24,36]. Additionally, the joint angles of the paretic lower limb and walking speed during assisted gait were measured to confirm the influence of kinematic factors on the muscle activity and intermuscular coherence of the VM.

### 2.5. Data Analysis

The joint angles of the paretic lower limb were determined by identifying the heel contact time of the paralytic side in each of the five gait cycles [37] using video, and the hip flexion angle was calculated using image analysis software (ImageJ, version 1.52a). The hip flexion angle was defined as the angle formed by the line parallel to the trunk and the line connecting the greater trochanter to the lateral condyle of the knee joint. The gait speed was calculated as the speed of walking on a 10-m walking path (with a 1-m runway). For the VM muscle activity, the early stance phase (heel contact-mid stance phase) of the paretic side was identified from the video, and the distal values were selected. The raw EMG signals were zero-lag 4th-order Butterworth filter and 5–450 Hz bandpass filtered, then mean-subtracted and rectified (in %). All EMG preprocessing was performed using the “Surface Electromyography for all EMG preprocessing was performed in accordance with the guidelines of “Surface Electromyography for the noninvasive Assessment of Muscles” (http://www.seniam.org accessed on 20 January 2010).

EMG–EMG coherence (intermuscular coherence) analysis was performed on two time series signals recorded from the proximal and distal portions of the VM. Amplitude squared coherence analysis (Welch) was performed on the tuning rate of two different time series signals for each frequency band; in the case of EMG, the coherence value appeared at 15–30 Hz (beta band) reflects the CST excitability [15,17]. Intermuscular coherence analysis is performed on full-wave rectified data, and this method increases the test-to-test reproducibility and reliability of variables derived from intramuscular coherence [17,38]. Data segments of 300 ms after heel contact on the paretic lower limb during walking were extracted from each cycle and then connected [39]. To reduce spectral leakage, the connected EMG signals were subjected to a Hamming window (window: 300, overlap: 150). The coherence between the two connected EMG signals (x and y) was defined as the square of the cross-spectrum normalized by the auto spectrum according to the following equation:$$C_{\mathrm{xy}}\left(f\right)=\frac{|P_{\mathrm{xy}}\left(f\right){|}^{2}}{P_{\mathrm{xx}}\left(f\right)P_{\mathrm{yy}}\left(f\right)}$$
where C_xy_ denotes the amplitude squared coherence for a given frequency (f). P_xx_(f) and P_yy_(f) indicate the x and y power spectra, respectively, and P_xy_(f) is the value of the cross-spectrum. The coherence function is the criterion for a linear correlation in the frequency domain and is output in the range of 0 to 1, where 1 indicates a perfect linear correlation. The intermuscular coherence estimate is the fraction of the activity of one surface EMG signal at a given frequency that can be predicted by the activity of the other surface EMG signal and quantifies the strength and frequency range of common synaptic inputs distributed across the motor neuron pool of the spinal cord. Since the coherence of the β-band (15–30 Hz) reflects CST excitability, we calculated the mean value of the β-band in this study [15,17].

The amount of immediate change in lower limb motor function on the paretic side due to differences in stimulation positions was calculated by dividing the values of the muscle activity and intermuscular coherence of VM at 20 min post-training by the values before training. The amount of each change in the motor function of the paretic lower limb in periods A and B was calculated by removing the trend by calculating the slope from the values of the muscle activity and intermuscular coherence of the VM at five time points, including the periods A and B. The trend was removed to correct for the effects of spontaneous recovery after stroke. MATLAB R2019b (MathWorks, Inc., Natick, MA, USA) was used for all data analyses.

## 3. Results

The subject did not experience any adverse effects during or after the experiment using tDCS.

### 3.1. Results of Clinical Assessments at the Period A End Time (Period B Start Time) and Period B End Time

The BRS score was I at the period A end time and I at the period B end time. The FMS score was 0 at the period A end time and 0 at the period B end time. The TIS score was 0 at the period A end time and 2 at the period B end time, and the FIM score for transfer was 2 at the period A end time and 3 at the period B end time (Table 1).

### 3.2. 20-Minute Short-Term Effects of Different Stimulation Positions of tDCS on Paretic Lower Limb Motor Function

The joint angle of the paretic lower limb (°, listed in order of pre-training/20 min post-training) was 26.1/26.5 for the no-tDCS condition, 28.0/26.1 for the uni-tDCS condition, and 25.9/26.8 for the bi-tDCS condition. Gait speed (m/sec, listed in order of pre-training/20 min post-training) was 0.38/0.37 for the no-tDCS condition, 0.38/0.38, for the uni-tDCS condition, and 0.39/0.40 for the bi-tDCS condition, and there was no significant change in both lower limb joint angle and gait speed. Figure 5 shows a typical example of the angular velocity (A) of the anterior-posterior tilt of the paretic lower leg during gait and the waveforms of the muscle activity of the VM in the proximal (B) and distal (C) parts during a gait cycle and the waveform of the intermuscular coherence (D) of the VM in each frequency band at 300 ms after heel contact. Next, the immediate changes in the muscle activity and intermuscular coherence of the VM due to the different stimulation positions of tDCS (the values after training divided by the values before training are described) are shown in Figure 6. The changes in the muscle activity were 1.0 for the no-tDCS condition, 1.0 for uni-tDCS condition, and 1.2 for bi-tDCS condition, respectively. The changes in the intermuscular coherence were 1.0 for the no-tDCS condition, 1.1 for uni-tDCS condition, and 1.2 for the bi-tDCS condition, respectively.

### 3.3. Effects of 4-Weeks Bi-tDCS Intervention on Paretic Lower Limb Motor Function

The joint angle of the paretic lower limb (°, Listed in order of intervention start time/intervention mid-time/intervention end time) was 27.6°/27.2°/26.5° in period A and 26.5°/27.0°/26.3° in period B, and gait speed (m/s) was 0.37°/0.37°/0.40° in period A and 0.40°/0.39°/0.43° in period B. The time series of the muscle activity and intermuscular coherence of VM at the five time points of periods A and B are shown in Figure 7A,C. The respective sums of the 4-weeks changes in the muscle activity and intermuscular coherence of the VM in periods A and B are shown in Figure 7B,D. Positive values of this change indicate greater improvement with training. The muscle activity of the VM was −10.94 in period A and 9.2 in period B and increased in period B. The intermuscular coherence of VM was −0.95 (×10^−1^) in period A and 0.52 (×10^−1^) in period B and increased in period B.

## 4. Discussion

In this study, we examined the effects of 20-min and 4-weeks bi-tDCS interventions on muscle activity and CST excitability of the paretic VM in a stroke patient with severe motor paralysis. As a result, the 20-min bi-tDCS intervention immediately improved muscle activity and inter-muscular coherence of the paretic VM compared to no-tDCS and uni-tDCS conditions. Furthermore, the 4-weeks bi-tDCS intervention also increased muscle activity and inter-muscular coherence of the paretic VM during assisted walking.

The muscle activity and intermuscular coherence of the VM are affected by the paretic lower limb joint angle and walking speed [40,41,42]. However, the paretic lower limb joint angle and gait speed observed in this study did not change among the conditions [37,43]. Therefore, we consider that the influence of kinematic factors on the muscle activity and intermuscular coherence of the VM is small. Next, in post-stroke patients who cannot walk independently, gait training [44] and tDCS on the primary motor areas of the injured hemisphere [18] increased motor-related area excitability and CST excitability in the injured side. However, in this patient, the uni-tDCS condition did not immediately change the intermuscular coherence of the VM compared to the no-tDCS condition. We consider this difference in the results to be due to severe motor paralysis. The subject in these studies had 30.9 ± 2.7 points for the lower limb subscale of the Fugl-Meyer-Assessment [44] and 4 points for the Brunnstrom Recovery Stage of the lower extremity [18]. However, this patient had a major injury to the coronet and posterior limb of the internal capsule, through which the CST passes, and had poor motor function, with 1 point for a BRS, 0 points for an FMS and 1 point for an FIM transfer scale for the start of period A. Previous studies reported that, in post-stroke patients with severe motor paralysis, there are limitations in increasing the activity of motor-related areas in the injured hemisphere and CST excitability from the injured hemisphere to the paretic lower limb muscles [6,8]. Therefore, it is likely that the uni-tDCS condition was not sufficient to increase the intermuscular coherence of VM, resulting in no increase in the muscle activity of the VM. 

Interestingly, in this patient with severe motor paralysis, the 20-min and 4-weeks bi-tDCS interventions increased muscle activity and intermuscular coherence in the VM. In patients with mild CST injury and motor paralysis, the CST excitability output from motor-related areas of the injured hemisphere to the paretic lower limb muscles during paretic lower limb movements influence the recovery of paretic lower limb motor function [6,7,45]. However, motor-related area excitability of the non-injured hemisphere and CST excitability from motor-related areas of the non-injured hemisphere to the paretic lower limb muscles were increased in post-stroke patients with severe motor paralysis [6,7,8,9,10]. This is because, in post-stroke patients with severe motor paralysis, motor-related areas, mainly the SMA and premotor areas of the non-injured hemisphere [6,7], and CST from the non-injured side to the paretic lower limb muscles [8,9,10] were selected as compensatory pathways. Additionally, for post-stroke patients who have difficulty walking independently, SMA excitability of the non-injured hemisphere increases during gait [23]. The previous study did not measure CST excitability, which is not entirely consistent with this study. However, these increases in excitability indicate the need to promote CST excitability output from motor-related areas, mainly SMA of the bilateral hemispheres, not just the injured hemisphere, as this is an important process for the recovery of paretic lower limb motor function and gait ability in post-stroke patients with severe motor paralysis. Therefore, we believe that assisted gait training combined with bi-tDCS increased the intermuscular coherence of the VM through SMA of the non-injured hemisphere. In addition, as a result of increased intermuscular coherence of VM, muscle activity was also increased, which is considered to be a study strength. The results of this study indicate the effectiveness of tDCS, including SMA of the non-injured hemisphere, in the rehabilitation strategy of post-stroke patients with severe motor paralysis.

This study had some limitations. First, KAFO has the potential to affect the magnitude of muscle activity in the VM because it immobilizes the knee joint in the extended position. In this patient, it was difficult to support the knee joint without immobilizing it in the extended position using KAFO, and this effect cannot be ruled out. However, considering that the waveform of the muscle activity of the paretic VM was confirmed to appear even during assisted walking with KAFO and that it was performed using the same orthosis, it is not considered a major problem. Second, in this study, intermuscular coherence was calculated only from the paretic VM. Therefore, the results may be different if intermuscular coherence is measured in other muscles. However, since the VM is an essential muscle that supports body weight, this result is beneficial for patients who lack support in their lower limbs, such as those with severe motor paralysis. Third, since TMS was not used in this study, we are unable to confirm whether the electrodes of the tDCS were optimally positioned to target the VM. We also have not been able to confirm whether uni-tDCS stimulated only the injured hemisphere. Fourth, we measured the CST excitability by tDCS to SMA, but did not confirm the excitability of the cortex itself. Finally, this study did not provide a sufficient period of washout or sham stimulation. However, based on the results in Figure 6 and Figure 7, it is highly likely that the effect was higher during the bi-tDCS period.

## 5. Conclusions

TDCS to bilateral SMA combined with gait training may increase the excitability of the CST and muscle strength of the paretic leg in a severe case of motor paralysis. It also provides the importance of increasing the excitability of SMA in the non-injured hemisphere in addition to the injured hemisphere.

## Figures and Tables

**Figure 1 brainsci-12-00452-f001:**
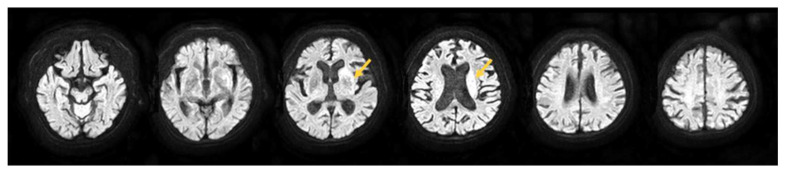
MRI showed a high-signal response in a wide area centered on the corona radiata and posterior leg of the internal capsule. Abbreviations: MRI, magnetic resonance imaging.

**Figure 2 brainsci-12-00452-f002:**
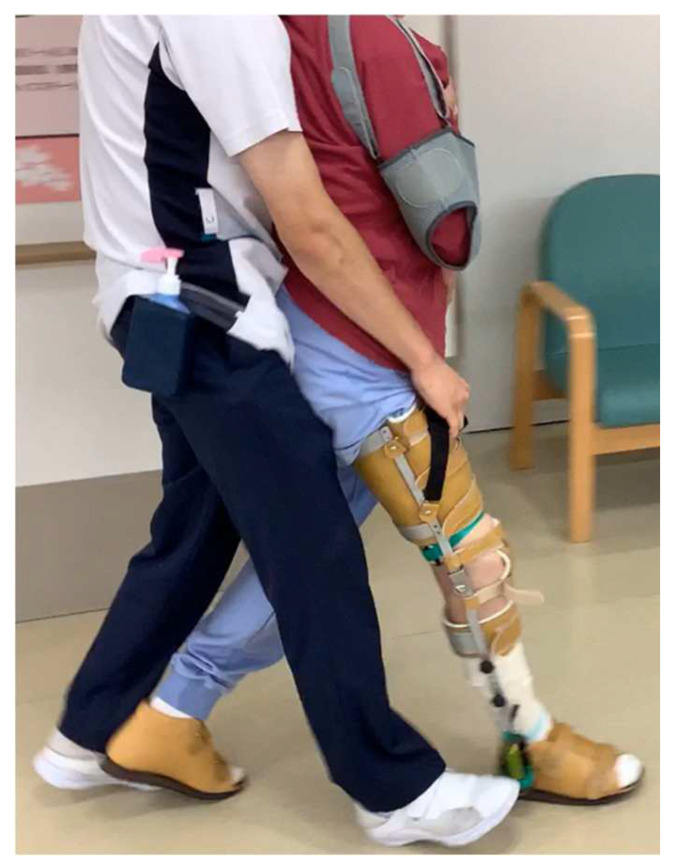
The therapist assists the patient in walking from the back using the KAFO. The right side shows the paralyzed side. Abbreviations: KAFO, knee-ankle foot orthosis.

**Figure 3 brainsci-12-00452-f003:**
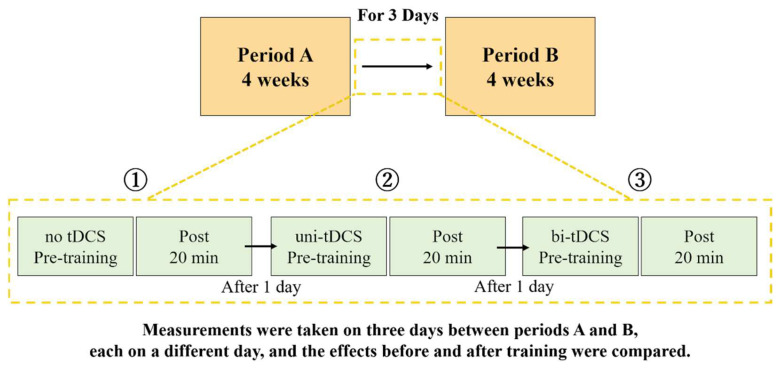
Assisted gait training only (no-tDCS) was conducted in period A, and tDCS over the SMA of the bilateral hemispheres combined with assisted walking training (bi-tDCS) in period B were conducted for 4 weeks each. The immediate effects of the three conditions (**1**) to (**3**) on the motor function of the paretic lower limb were measured before and 20 min after training in the period between periods A and B (3 days). Abbreviations: tDCS, transcranial direct current electrical stimulation.

**Figure 4 brainsci-12-00452-f004:**
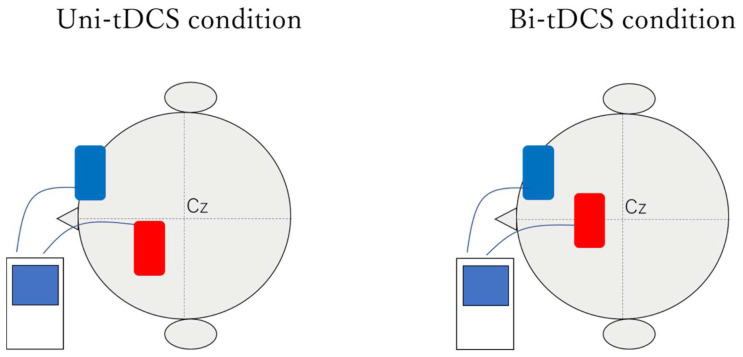
The red electrode in the figure indicates the anode and the blue indicates the cathode. The anode was placed in the SMA of the injured hemisphere (3 cm anterior to the lateral side of Cz) for the uni-tDCS condition and in the SMA of the bilateral hemispheres (3 cm anterior to Cz) for the bi-tDCS condition based on the International Electroencephalogram 10–20 method. The cathode was placed in the supraorbital region. Abbreviations: SMA, supplementary motor area; tDCS, transcranial direct current electrical stimulation.

**Figure 5 brainsci-12-00452-f005:**
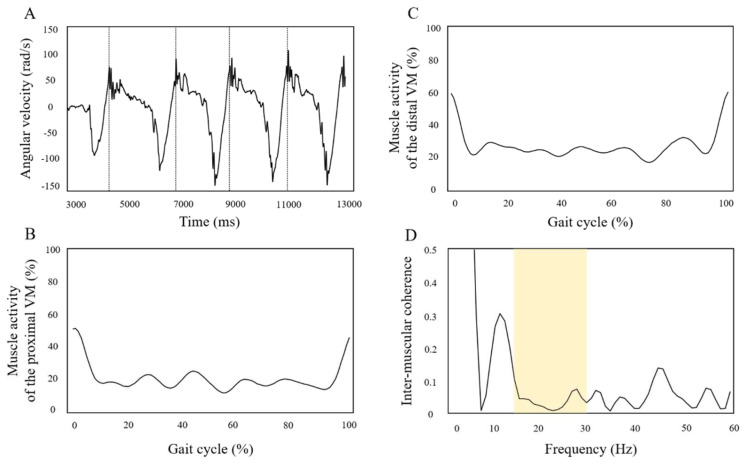
The angular velocity of the anteroposterior tilt of the paretic lower leg during gait is shown in (**A**). The positive values in the graph indicate the anterior tilt of the lower leg, and the dashed line indicates the time of heel-ground contact. The muscle activities of the VM in the proximal (**B**) and distal (**C**) parts during the gait cycle are shown. The 0 percent of the gait cycle is the timing of heel-ground contact on the paretic side. (**D**) shows the intermuscular coherence of the VM. The yellow box indicates the 15–30 Hz range.

**Figure 6 brainsci-12-00452-f006:**
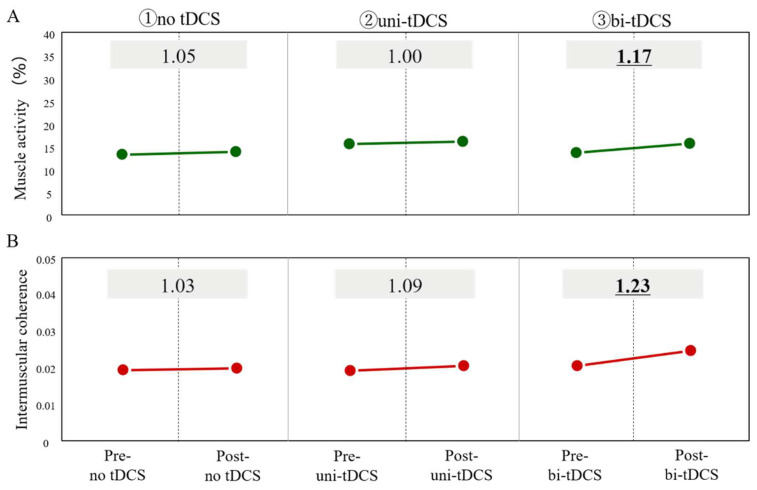
Immediate changes in the muscle activity (**A**) and intermuscular coherence (**B**) of the VM due to the different stimulation positions of tDCS are shown. The values in the graphs are the post-training values divided by the pre-training values. Abbreviations: tDCS, transcranial direct current electrical stimulation.

**Figure 7 brainsci-12-00452-f007:**
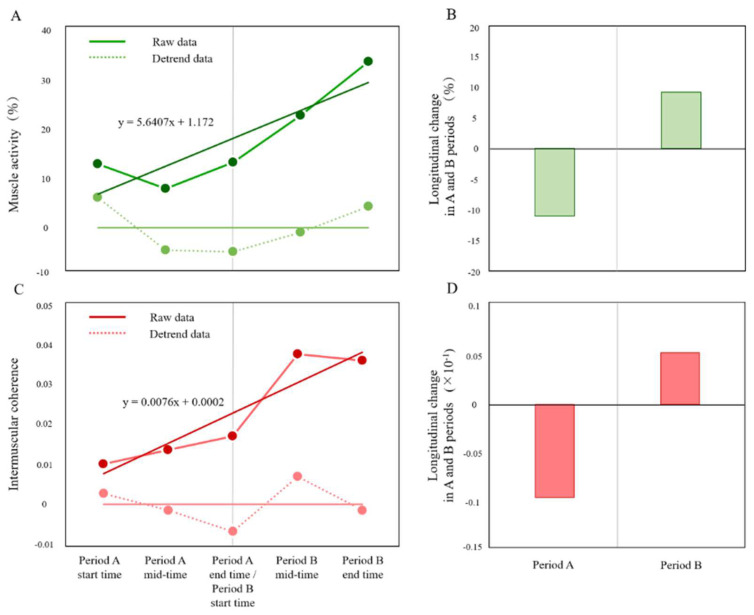
(**A**,**C**) shows the muscle activity and intermuscular coherence of the paretic VM at the five time points of periods A and B combined. The dark green and dark red lines show the raw data, and the light green and light red dashed lines indicate the detrended data. (**B**,**D**) shows the sum of respective 4-weeks changes in the muscle activity and intermuscular coherence of the paretic VM in periods A and B.

**Table 1 brainsci-12-00452-t001:** Patients’ clinical characteristics.

	Period A Start Time	Period A End Time	Period B Start Time	Period B End Time
BRS (lower limb): max = 6	Ⅰ	Ⅰ	Ⅰ	Ⅰ
FMS (lower limb): max = 22	0	0	0	0
TIS: max = 23	0	0	0	2
FIM for transfer: max = 7	1	2	2	3

Abbreviations: BRS, Brunnstrom Recovery Stage; FMS, Fugl-Meyer Assessment synergy item; TIS, Trunk Impairment Scale; FIM, Functional Independence Measure.

## Data Availability

The datasets used and/or analyzed during the current study are available from the corresponding author on reasonable request.

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
