# Peer review of "Effects of Transcranial Direct Current Stimulation of Bilateral Supplementary Motor Area on the Lower Limb Motor Function in a Stroke Patient with Severe Motor Paralysis: A Case Study"

_brainsci, 2022, doi:10.3390/brainsci12040452_

Round 1

Reviewer 1 Report

The authors present a case report where the patient (30 days post-stroke at time of enrollment) participated in two treatment phases A (no tDCS) and B (tDCS) for 4 weeks each. Additional, the patient underwent a cross-over design between session A and B, where the immediate effects of Uni-, Bi- and No- tDCS were tested on muscle activity and coherence. Overall, they report that only “Bi-tDCS” demonstrated immediate and longitudinally increase muscle activity and coherence.

Overall, the background, methods and results are over-stated and the methodology is not adequate in order to make any meaningful conclusions.

Background: The author makes strong statements without supporting citations. For example, in the introduction the authors state that “It has been shown that tDCS to the motor-related areas of the injured hemisphere 45 restores the motor function of the paretic leg in stroke patients [12,14–17].” Restoration is a very strong in inaccurate phrasing of the effect of tDCS in general. Furthermore, the work that was cited are not relevant and/or are not adequate to support this premise (e.g. Nitsche et al was an able-bodied TMS study, Dutta is a conference proceeding, Tanaka et al is a single session observation of force production, Jefferey et al is an able-bodied TMS study).
There is no introduction why the authors chose “SMA” placement rather than “M1” placement of the tDCS electrodes. Frankly, given the non-specific nature of tDCS, the difference between SMA and M1may be non-substantial, but still is relevant given the authors general topic of discussion. Based on a citation use in the manuscript, tDCS has been placed over the TA area of the precentral gyrus using TMS as a guide (Chang et al 2015).

Methods and Results: In terms of the placement of the electrodes, a true ‘Unilateral’ approach will certainly be difficult to achieve without the aid of TMS given the close proximity of the lower-limb to the longitudinal fissure. Their ‘null’ findings with the “unilateral” may simply be the result of not adequately targeting the lower limb. Given that this is a case report and TMS was not used to identify targets; the manuscript would benefit from current modeling – especially considering that the same ‘reference’ electrode was used in both locations. Even with the ‘unilateral’ approach, the current still flows between each hemisphere – still it’s unclear if the current density sufficiently targeted the lower-limb in the unilat montage (and even the bilat montage) given the difference in trajectory between the active and reference electrodes between the ‘bilat’ and the ‘unilat’ montage. Further, when the patient received “no-tDCS” was a sham protocol used? This is particularly important given that this is a case report and the participant was not naïve to active-TDCS.

It’s difficult to draw any meaningful conclusions between the A or B 4-week treatment phase, given that the patient was still likely undergoing spontaneous recovery (enrolled 30-days post-stroke). Given that the B active –tDCS session was after the A session, it’s not clear if any effects were due to tDCS or spontaneous recovery. I suspect a similar result would have been found if the sessions were flipped (i.e. active tDCS first and no-tDCS second). Likewise, without adequate sham-stimulation it’s unknown if the patient was more motivated given the ‘experimental treatment’ during session B. In terms of the cross-over design, without any indication as to the reliability of coherence and muscle activity, it’s difficult to draw any meaningful conclusions in a case-report.

Conclusion: “TDCS to bilateral motor-related areas combined with Gait training is effective in restoring paretic lower limb motor function in severe cases of motor paralysis, and increasing excitation of motor-related areas in the non-injured hemisphere in addition to the injured hemisphere is useful.” This is a very strong and unsupported statement given the aforementioned comments on methodology as well as the general nature of a case report.

Author Response

Thank you very much for your valuable comments. Please refer to the attached file for our response to what you have pointed out.

Reviewer 2 Report

Thank you very much for the opportunity to review the manuscript.

The study I believe is of interest to the scientific community. I think it would benefit from using the "Case Report Guidelines" document as a reference.

Abstract
I think the abstract is well written and contains all relevant information given that it complies with items 3a, 3b, 3c, and 3d of the "Case Report Guidelines".

Keywords: I suggest including the words "case report" as advised in item 2 of the "Case Report Guidelines".

Introduction
I think it is adequate since it indicates why this case is unique and includes relevant references and some of them are from the year 2021.

Patient information

I suggest the incorporation of the family and psychosocial history, following item 5b of the "Case Report Guidelines".

Design
Figures 2 and 3 I believe are of interest and help us to adequately understand the intervention, as well as to provide a timeline of the intervention.

Were variables that can affect transcranial stimulation controlled in any way? I recommend using Table 1 in the manuscript: Thair, H., Holloway, A. L., Newport, R., & Smith, A. D. (2017). Transcranial direct current stimulation (tDCS): a beginner's guide for design and implementation. Frontiers in neuroscience, 11, 641. to describe whether different variables that can affect transcranial direct current stimulation were controlled. In case some of the variables were not controlled this should be included in the limitations of the study.

Results
I believe the results are well written and adequately understood.

Discussion
I think it addresses the fundamental points, as well as includes relevant and up-to-date limitations and citations.

Conclusion
I believe that the conclusion is not adequate given that with a case study one cannot make a categorical statement about the effectiveness of an intervention. The conclusion should be reworded.

When resubmitting the manuscript please attach the CARE checklist of information that should be included when writing a case report (https://www.care-statement.org/checklist), with the line in the manuscript where each item is included.

Author Response

(The authors gave the same response as above.)

Reviewer 3 Report

  • The introduction is missing neurophysiological techniques that can be used to evaluate the integrity of CST (such as nTMS), recording motor evoked potentials, or using imaging techniques. Please update accordingly with references.

-please use consistently through the manuscript the acronym CST, for example row 58, 62.

  • Please use the acronyms consistently, look for Table 1 the functional scoring tests are not provided in the acronym, but previously in the text, these acronyms were provided (rows 71-74). For example row 266 full term for Fugel Meyer Assessment was given, but acronym was introduced previously rows 71-74.
  • Row 79, if the paper is on the single patient then this sentence needs to be adjusted “Written informed consent was obtained from all patients. The same is with the sentence “The subjects did not study any adverse effects during or after the experiment using tDCS.”
  • It would be appreciated if authors could use TMS (nTMS) to evaluate integrity of CST on both sides
  • Fig 4 please indicate colors red and blue to know for anode vs cathode electrode as legend
  • It is suggested to shortly present the use and interpretation of the FMS, TIS, and FIM. Is one person/expert scored the patient or do you have two examiners? Was the examiner blinded?
  • The score for Brunnstrom Recovery Stage of the lower extremity is presented in the Discussion section, but not in the methodology and results section
  • VS coherence longitudinally increased, please present more understandably to the readers the term “longitudinally” what does it mean? 1day? Paragraph 3.3 and in the discussion section.
  • Please write more clearly or rewrite the sentence (row 278-280)

“In patients with mild CST injury and motor paralysis, the excitability of the 278

SMA and premotor cortex in the injured hemisphere during paretic lower limb movements influence the recovery of paretic lower limb motor function [6,7,38], and it is not

important in the non-injured hemisphere”. “and it is not

important in the non-injured hemisphere” what this means, is not understandable.

-If SMA was supposed to be stimulated, why using the term “motor-related areas”? Why not simply use the term SMA in the whole manuscript, and also change the title, such as: “Effects of transcranial direct current stimulation of the unilateral and bilateral supplementary motor area on the lower limb motor function in a stroke patient with severe motor paralysis: a case study”

-row 281-283 the authors refer to studies measuring cortical excitability but should be more clear in disentangling their results from these studies since the submitted work did not apply cortical excitability measurement.

-The limitation of the study to be included is not conducting motor cortical excitability measurement, non-conducting assessment of the integrity of corticospinal tract with a functional test like motor evoked potentials with TMS.

Author Response

(The authors gave the same response as above.)

Round 2

Reviewer 2 Report

The authors have made the requested changes. Please include in material and method that the CARE guidelines (for CAse REports) have been followed.

Author Response

Thank you very much for reviewing our manuscript. We revised the manuscript based on your suggestions. Thanks to you, we believe that the methodology of this study is now clearer. Please refer to the file for the response letter.

Reviewer 3 Report

Thank you for the provided answers. Please check the grammatic again in the whole manuscript (i.e. row 357 "Gait" should be with a small letter).